# Variable Inhibition of DNA Unwinding Rates Catalyzed by the SARS-CoV-2 Helicase Nsp13 by Structurally Distinct Single DNA Lesions

**DOI:** 10.3390/ijms25147930

**Published:** 2024-07-19

**Authors:** Ana H. Sales, Iwen Fu, Alexander Durandin, Sam Ciervo, Tania J. Lupoli, Vladimir Shafirovich, Suse Broyde, Nicholas E. Geacintov

**Affiliations:** 1Chemistry Department, New York University, 31 Washington Place, New York, NY 10003, USA; ah6@nyu.edu (A.H.S.); ad8@nyu.edu (A.D.); src536@nyu.edu (S.C.); tjl229@nyu.edu (T.J.L.); vs5@nyu.edu (V.S.); 2Biology Department, New York University, 24 Waverly Place, New York, NY 10003, USA; if539@nyu.edu (I.F.); broyde@nyu.edu (S.B.)

**Keywords:** SARS-CoV-2, Nsp13 helicase, SF1 helicase, unwinding, processivity, protein–DNA interactions, UV photolesions, crosslinked thymine dimer, CPD, 6–4PP, benzo[*a*]pyrene diol epoxide–DNA adducts

## Abstract

The SARS-CoV-2 helicase, non-structural protein 13 (Nsp13), plays an essential role in viral replication, translocating in the 5′ → 3′ direction as it unwinds double-stranded RNA/DNA. We investigated the impact of structurally distinct DNA lesions on DNA unwinding catalyzed by Nsp13. The selected lesions include two benzo[*a*]pyrene (B[*a*]P)-derived dG adducts, the UV-induced cyclobutane pyrimidine dimer (CPD), and the pyrimidine (6–4) pyrimidone (6–4PP) photolesion. The experimentally observed unwinding rate constants (*k*_obs_) and processivities (*P*) were examined. Relative to undamaged DNA, the *k*_obs_ values were diminished by factors of up to ~15 for B[*a*]P adducts but only by factors of ~2–5 for photolesions. A minor-groove-oriented B[*a*]P adduct showed the smallest impact on *P*, which decreased by ~11% compared to unmodified DNA, while an intercalated one reduced *P* by ~67%. However, the photolesions showed a greater impact on the processivities; notably, the CPD, with the highest *k*_obs_ value, exhibited the lowest *P*, which was reduced by ~90%. Our findings thus show that DNA unwinding efficiencies are lesion-dependent and most strongly inhibited by the CPD, leading to the conclusion that processivity is a better measure of DNA lesions’ inhibitory effects than unwinding rate constants.

## 1. Introduction

Helicases play critical roles in the maintenance of genome stability, including DNA replication, transcription, and repair [1], as well as RNA metabolism, such as splicing and ribosome assembly [2]. The lifecycle of the SARS-CoV-2 virus and its replication in infected human host cells depends critically on its mechanism of replication. Non-structural protein 13 (Nsp13), an RNA helicase, belongs to the helicase superfamily 1 (SF1) and plays a critical role in viral replication; it first unwinds double-stranded RNA to provide a single-stranded template for the subsequent transcription of the viral genome. The COVID-19 pandemic stimulated significant interest in the design of new inhibitors [3,4,5,6,7,8,9] to suppress SARS-CoV-2 Nsp13 helicase unwinding activities and subsequent viral replication and proliferation [10,11]. The Nsp13 helicase unwinds undamaged double-stranded RNA (dsRNA) [10,12] and DNA (dsDNA) [10,12,13,14,15,16], with similar rates and efficiencies, using an ATP-driven 5′ → 3′ polarity translocation mechanism. The helicase can be inhibited either by blocking ATP hydrolysis [12] or by inhibiting the unwinding of the Nsp13 helicase without affecting ATPase activity [15,17,18,19,20,21].

Significant efforts have been invested to identify effective inhibitors of the ATPase activity of Nsp13 [4,6,8,11,22,23,24]. The impact of the non-covalent binding of various small molecules on the unwinding of double-stranded DNA by various helicases has been studied [25]. However, non-covalent inhibitor–DNA binding can include intercalative complexes and/or external minor or major groove DNA binding sites in unknown proportions. Examples of the effects of various chemicals on the unwinding of double-stranded DNA have been previously described [26,27]. 

In order to obtain new and mechanistic structure–function relationships, stable covalent DNA lesions with known DNA conformations are well suited for investigating DNA lesion structure–helicase unwinding relationships. Indeed, it has been demonstrated that covalently attached DNA adducts, including adducts derived from polycyclic aromatic hydrocarbons (PAHs), slow the progress of the human 3′ → 5′ helicases WRN [27] and RecQ [28]. 

The objective of this work was to gain a new understanding of how the chemical structures and physical conformations of different DNA lesions affect DNA unwinding catalyzed by the Nsp13 helicase. We focused on the Nsp13-catalyzed unwinding of double-stranded DNA containing single UV-induced DNA photolesions and single benzo[*a*]pyrene-diol epoxide (B[*a*]P) reaction products that, in vivo or in vitro, form the bulky B[*a*]P-*N*^2^-dG (guanine) adducts shown in Figure 1. The designation ‘bulky’ refers to the physical dimensions of the B[*a*]P aromatic ring system. These adducts are known from previous studies to inhibit helicases [27,28,29], including Nsp13.

We embedded these lesions into dsDNA (Figure 2) and determined their impact on Nsp13-catalyzed unwinding, focusing on the overall unwinding rate constants *k*_obs_ and the processivities *P*. The UV-induced DNA lesions selected in our current study include UV radiation-induced crosslinked thymine (T) dimer lesions, the *cis-syn* cyclobutane pyrimidine dimer (CPD), and the pyrimidine (6–4) pyrimidone photoproduct (6–4PP) (Figure 1A). The CPD lesion has two open book-like thymine bases that are covalently crosslinked via bonds between the two C5 and C6 atoms. The 6–4PP lesion has two nearly perpendicular thymine rings linked by a single C6–C4 covalent bond. NMR studies showed that these two lesions are structurally very distinct [30,31]: 6–4PP is more distorting than the CPD, since the hydrogen bonding at the 6–4PP lesion site is ruptured, and the helix is destabilized and bent by ~44°. By contrast, in the case of the CPD, Watson–Crick hydrogen bonding is minimally perturbed, and the helix is bent by only ~9° relative to the unmodified duplex. Similar features of hydrogen bonding at the lesion site are also revealed in the crystal structures of nucleosomes containing a CPD (PDB ID 5B24 [32]) or 6–4PP lesion (PDB ID 4YM6 [33]) (Figure 1A). 

The PAH adducts investigated here include two stereoisomeric benzo[*a*]pyrene diol epoxide-derived (B[*a*]P) *N*^2^-deoxyguanosine adducts: the (+)-*trans*-B[*a*]P-dG (*trans* adduct) and the (+)-*cis*-B[*a*]P-dG adduct (*cis* adduct) (Figure 1B). The structural features of these two stereoisomeric B[*a*]P-*N*^2^-dG adducts positioned in double-stranded DNA duplexes have been established by high-resolution NMR methods (Figure 1B) [34,35]. The *trans* adduct has the B[*a*]P aromatic ring system positioned externally in the minor groove of DNA oriented toward the 5′-direction of the modified strand [34]. The *cis* adduct has the base-displaced intercalated conformation; the B[*a*]P aromatic ring system is intercalated between adjacent base pairs, and the modified guanine base is displaced into the minor groove, while its partner base cytosine (C) is displaced into the major groove [35] (Figure 1B).

**Figure 1 ijms-25-07930-f001:**
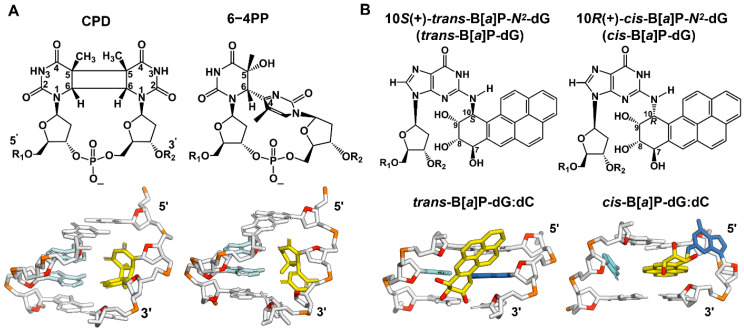
(**A**) The chemical and crystal structures of the CPD and 6–4PP. Their structural features in the DNA duplexes are shown; the thymine dimers (yellow) are paired with adenines (cyan) in the opposite strand. Note that the 4-mer DNA duplexes are taken from the crystal structures of a nucleosome containing a CPD (PDB ID 5B24 [32]) or 6–4PP lesion (PDB ID 4YM6 [33]). (**B**) Stereoisomeric 10S (+)-*trans*-B[*a*]P-*N*^2^-dG (*trans*) and 10R (+)-*cis*-B[*a*]P-*N*^2^-dG (*cis*) adducts. Their structural features in the 3-mer DNA duplex are shown. The B[*a*]P-modified-*N*^2^-guanine (blue) residues are paired with cytosine (cyan) in the opposite strand. The B[*a*]P-derived DNA adducts are designated as ‘bulky’ because they contain five six-carbon rings (carbons in yellow and oxygens in red), while the non-bulky CPD and 6-4 contain none.

**Figure 2 ijms-25-07930-f002:**
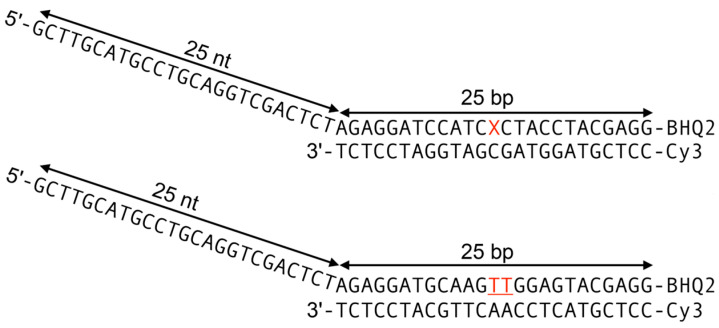
Definitions of the DNA substrates. The B[*a*]P lesions and the thymine dimers, denoted by the red X and TT, respectively, are positioned in the double-stranded region of the upper strand.

At the 5 nM DNA concentration employed in these experiments (similar to our previous study of the RecQ helicase [28]), the re-association rate of two separated strands was negligible on the time scale of the experiment. Prior to the unwinding experiments, the DNA substrates were pre-equilibrated with the Nsp13 protein (60 nM), and the unwinding reactions were initiated by mixing this solution with aliquots of the unwinding buffer, containing 20 mM TRIS-HCl, pH 7.6, 10 mM KCl, 5 mM MgCl_2_, 2 mM dithiothreitol (DTT), 5% glycerol, 0.1 μg/μL bovine serum albumin (BSA), and 2 mM ATP. All experiments were conducted at 25 °C. 

Our kinetic analyses show that *k*_obs_, which is defined as (*k*_U_ + *k*_D_), and the processivity *P* = *k*_U_/(*k*_U_ + *k*_D_) vary depending on the nature of the DNA lesion (*k*_U_, unwinding rate constant; *k*_D_, helicase–DNA dissociation rate constant). The B[*a*]P DNA adducts exhibit the greatest impact on the *k*_obs_ values relative to the unmodified DNA and the smallest impact on the processivity (*P* = 0.89) for the *trans* adduct, while the photolesions exhibit the lowest processivity (*P* = 0.11) for the CPD. The processivity is a better measure of the inhibitory effects of DNA lesions than the unwinding rate constant. Thus, our findings indicate that the DNA unwinding efficiencies are lesion-dependent and most strongly inhibited by the CPD, highlighting the different impacts of structurally distinct DNA lesions on the helicase activity. These observations provide valuable insights into the relationships between the covalent DNA adduct structures and the duplex unwinding catalyzed by Nsp13. This study offers a unique new understanding of the mechanism of Nsp13 helicase activity as it processes our structurally distinct DNA lesions. 

## 2. Results

### 2.1. Nsp13-Catalyzed Unwinding Phenomena

When ATP is added to a solution of unmodified DNA, a rapid burst in fluorescence intensity is observed that is attributed to pre-existing Nsp13-DNA complexes (Figure 3). The burst is followed by a slower increase in fluorescence intensity that is associated with the diffusion-dependent formation of bimolecular Nsp13-DNA complexes, which causes the unwinding of the double-stranded DNA region, as reported by other workers [36]. The unwinding of double-stranded DNA is initiated by the addition of ATP, which results in a step-by-step unwinding process with a step size of 9.3 base pairs in the case of unmodified DNA [15,37]. The time dependence of the unwinding of double-stranded DNA by helicases can be described by exponential growth curves, macroscopic unwinding rate constants *k*_obs_, and processivities *P* [36,37]:*I*(*t*) = *I*_burst_ *+ P* [1 − exp(−*k*_obs_*t*)](1)

The time-dependent Cy3 fluorescence intensity *I*(*t*) is proportional to the fraction of unwound DNA. The *I*_burst_ term, denoting a burst of unwinding, represents the rapid DNA unwinding induced by pre-existing non-covalent helicase–DNA complexes. This rapid burst phase is followed by a slower phase due to subsequent complex formation and unwinding kinetics. The unwinding process is determined by the competition between the unwinding rate constant (*k*_U_) and the dissociation rate constant (*k*_D_) of the Nsp13-DNA complex. The best fit of Equation (1) to the experimentally measured DNA unwinding curve yields the magnitude of the overall unwinding rate *k*_obs_, which is defined as (*k*_U_ + *k*_D_), while the processivity *P* depends on the ratio of rate constants *k*_D_/*k*_U_:*P* = *k*_U_/(*k*_U_ + *k*_D_) = 1/(1 + *k*_D_*/k*_U_)(2)

The processivity *P* is defined as the probability that a helicase will unwind the DNA before it dissociates from the helicase–DNA complex during the unwinding reaction.

The analysis of the unmodified DNA unwinding curve (Figure 3) yields a *k*_obs_ value of (21.9 ± 0.03) × 10^−3^ s^−1^ at concentrations of 5 nM DNA substrate and 60 nM Nsp13. 

### 2.2. Impact of Nsp13 Helicase Concentration on DNA Unwinding 

Upon adding ATP to a solution of Nsp13 and unmodified double-stranded DNA constructs, the fluorescence intensity increases as the DNA molecules are unwound (Figure 3). The sharp rise in fluorescence intensity (burst) immediately after the addition of ATP is attributed to the prompt unwinding of pre-existing protein–DNA complexes. The burst accounts for 20–23% of the fully unwound fluorescence yield at a 60 nM Nsp13 concentration [Nsp]. The amplitude of the burst phase increases with increasing Nsp13 concentration in a manner consistent with the standard bimolecular binding equation:DNA_UW_ = [Nsp]/([Nsp] + *K*_D_)(3)

Here, DNA_UW_ represents the fraction of unwound DNA, and *K*_D_ is the equilibrium dissociation constant. The DNA was unwound by Nsp13 in a protein-concentration-dependent manner, regardless of the presence of lesions (Figure 4). The solid line in Figure 4 represents the best fit of Equation (3) to the experimental data points with the dissociation constant *K*_D_ = 28 nM. 

### 2.3. Unwinding Kinetics of DNA Substrates Containing DNA Lesions

#### 2.3.1. DNA Unwinding Curve for B[*a*]P Adducts

When B[*a*]P DNA adducts are present in the double-stranded region, bursts are not observed at the 60 nM Nsp13 concentration adopted in this study (Figure 5). (However, burst phases were observable at Nsp13 concentrations of ~180 nM and higher, but they were not further investigated here.) While it is generally noted that helicases bind to the single-stranded DNA and single-strand/double-strand (ss-ds) junction, the observed loss of the burst phase suggests that Nsp13 accesses the double-stranded region of ss-ds DNA sequences.

Analyses of the B[*a*]P DNA adducts’ duplex unwinding curves reveal that the *k*_obs_ values of the *trans* and *cis* adducts are (1.60 ± 0.02) × 10^−3^ s^−1^ and (1.47 ± 0.02) × 10^−3^ s^−1^, respectively (Figure 5). These values represent significant reductions of ~14-to-15-fold relative to the unmodified DNA with a *k*_obs_ value of (21.9 ± 0.03) × 10^−3^ s^−1^ at the same Nsp13 and DNA concentrations.

#### 2.3.2. DNA Unwinding Curves for UV-Induced DNA Lesions

Figure 6 illustrates that when CPD or 6–4PP lesions are present in the double-stranded region, bursts are not seen at the same concentrations of Nsp13 and DNA used in unmodified DNA. The *k*_obs_ values determined from the 6–4PP and CPD unwinding curves are (4.02 ± 0.04) × 10^−3^ s^−1^ and (11.60 ± 0.04) × 10^−3^ s^−1^, respectively. Relative to the unmodified DNA, these *k*_obs_ values represent a ~50% reduction for CPD and an ~82% reduction for 6–4PP.

### 2.4. DNA Unwinding Kinetic Parameters 

In this work, we used kinetic methods to determine the overall unwinding rate constant *k*_obs_ using unmodified DNA as a benchmark. The experimentally observed DNA unwinding rate constants (*k*_obs_) and the kinetic parameters (*k*_U_, *k*_D_, *P*) associated with the different DNA lesions studied are summarized in Table 1. 

The *k*_obs_ values vary depending on the nature of the DNA lesions and are ~2-to-15-fold slower than in the case of undamaged DNA (Table 1). The *k*_obs_ values are diminished by similar factors of ~14–15 by the *trans* and *cis* adducts and only by factors of ~2–5 by the CPD and 6–4 photolesions. The overall unwinding rate *k*_obs_, which is defined as (*k*_U_ + *k*_D_), is determined by the competition between the unwinding rate constant (*k*_U_) and the dissociation rate constant (*k*_D_) of the Nsp13-DNA complex. These rate constants, *k*_U_ and *k*_D_, associated with different DNA lesions are depicted in Figure 7. As expected, the steric hindrance and local structural distortions caused by the DNA lesions would hinder the step-wise displacement of the helicase and thus diminish the unwinding rate constant *k*_U_. In turn, the residence time of the helicase at the site of the adduct is extended. However, this enhanced residence time might be correlated with an increase in the dissociation rate constant *k*_D_ of the Nsp13-DNA complex. 

Equations (1) and (2) predict that the processivity *P* depends on the *k*_D_*/k*_U_ ratio (Table 1), which defines the fraction of unwound DNA. Thus, helicase processivity *P* provides a measure of the fraction of unwound DNA before dissociation. Among all of the lesions, relative to the unmodified DNA with *P* = 1 (100% unwinding), the *trans* adduct exhibits the smallest impact on the value of processivity (*P* = 0.89), which decreased by ~11%, while the *cis* adduct reduces processivity (*P* = 0.33) by ~67% (Table 1). On the other hand, UV-induced DNA lesions show a greater impact on the processivities than the B[*a*]P DNA adducts. Strikingly, the CPD lesion, though it has the highest *k*_obs_ value, exhibits the lowest value of processivity (*P* = 0.11), diminished by as much as ~89%. Altogether, the increase in *k*_D_ and the decrease in *k*_U_ may combine in a way that may result in similar changes in the overall unwinding rate constant *k*_obs_. However, the *k*_D_*/k*_U_ ratio, which determines processivity, is a much more sensitive and accurate indicator of the inhibitory effects of DNA lesions.

## 3. Discussion

### 3.1. DNA Unwinding Kinetic Parameters Are Lesion-Dependent

Helicases are characterized by kinetic parameters that define their activity. In this work, we used kinetic methods to determine the overall unwinding rate constant *k*_obs_ using unmodified DNA as a benchmark. Our analyses reveal that *k*_obs_ and the kinetic parameters, including *k*_U_, *k*_D_, and *P*, vary depending on the nature of the DNA lesions (Table 1). These lesion-dependent kinetic parameters highlight the different impacts of the structurally distinct DNA lesions on the unwinding of double-stranded DNA catalyzed by the Nsp13 helicase. These observations provide valuable insights into the relationships between the covalent DNA adduct structures and the duplex unwinding catalyzed by Nsp13, discussed below, and provide further understanding of the mechanism of Nsp13 helicase activity. 

### 3.2. Impact of B[a]P Adducts on NSP13-Catalyzed Unwinding Kinetics 

The presence of B[*a*]P–DNA lesions in the double-stranded region of the modified DNA sequences (Figure 1B and Figure 2) results in a significant decrease in the overall unwinding rate constant *k*_obs_ to a similar extent for both *trans* and *cis* adducts, but there is a pronounced difference in their processivities (Table 1). These observations indicate that when compared to unmodified DNA, for the *trans* case, the decrease in *k*_obs_ is predominantly due to the decrease in the unwinding rate constant *k*_U_ (Figure 7), which is likely associated with steric hindrance effects. However, in the *cis* case, the decrease in *k*_obs_ is caused not only by a much slower unwinding rate constant *k*_U_ but also by a much faster dissociation rate constant *k*_D_ than in the *trans* adduct (Figure 7).

The distinct structural features of these B[*a*]P adducts in dsDNA provide insights into these observations. In particular, the *k*_D_*/k*_U_ ratio is much greater for the *cis* adduct than the *trans* adduct, resulting in a noticeable reduction in *P*. In the *trans* duplex, the bulky aromatic B[*a*]P ring system is located at an external minor groove DNA binding site, with the Watson–Crick pairing more or less intact at the lesion site [34]. However, the *cis* adduct adopts a base-displaced intercalated conformation [38], with a fully ruptured Watson–Crick pair at the lesion site due to the extrusion of both the modified base G and its partner base C out of the helix (Figure 1B). Notably, its planer aromatic B[*a*]P ring system, intercalated between adjacent base pairs, provides stabilizing π–π stacking interactions, which are not found in the *trans* adduct. Accordingly, the *cis* adduct is scarcely thermodynamically destabilized compared to the unmodified duplex (~4 °C), while the *trans* adduct is destabilized by ~10 °C [39]. Note that the degree of destabilization caused by the DNA lesions is reflected in the differences in 11-mer duplex melting points, ΔT_m_ = T_m_(modified duplex) − T_m_(unmodified duplex) [40]. Moreover, the *cis* adduct is more inhibitory because its planar aromatic ring system is oriented near-perpendicular relative to the direction of translocation of the helicase. These differences between the two adducts lead to the inference that the duplex with the *cis* adduct will resist unwinding by the helicase more than the one with the *trans* adduct, leading to a much slower unwinding rate constant *k*_U_ than in the *trans* adduct duplex (Figure 7). This slower unwinding rate *k*_U_ results in a longer residence time for the Nsp13 helicase bound to the site of the *cis* adduct; consequently, this increases the probability of dissociation of the Nsp13-DNA complex and leads to an increase in the *k*_D_ value. Furthermore, the expelled bases at the site of the *cis* adduct from the interior of the duplex DNA may inhibit some of the local Nsp13-DNA contacts, further facilitating the greater dissociation of Nsp13 from the *cis*-containing duplex than for the *trans*-containing duplex. Consequently, all of these effects combined in the case of the *cis* adduct cause a greater decrease in the processivity *P* than in the case of the *trans* adduct. 

Overall, the distinct conformational features in the duplex imposed by these two DNA adducts could explain why the *k*_D_/*k*_U_ ratio is much greater for the *cis* than the *trans* adduct and, thus, why the processivity of DNA unwinding is lower in the case of the *cis* adduct. These observations indicate that these adducts inhibit the Nsp13 helicase activity in a stereospecific manner. The *cis*-B[*a*]P-dG adduct is a significantly more effective inhibitor than the *trans*-B[*a*]P-dG adduct, indicating that Nsp13 unwinding is sensitive to adduct stereochemistry. Thus, it is likely that differences in the orientation of the *cis* and *trans* adducts relative to the attached guanine would affect translocation at the ss-ds junction when the B[*a*]P ring system is threaded through. As discussed elsewhere, the translocating strand bearing the B[*a*]P ring system must pass through a narrow channel for successful unwinding to occur [41]. Also, since the orientations of these adducts are governed rigorously by their stereochemistry, their conformations in single-stranded and double-stranded DNA manifest similar features [42,43]. 

### 3.3. Impact of the UV-Induced DNA Lesions on NSP13-Catalyzed Unwinding Kinetics

The behavior of the two UV photolesions is qualitatively different from that of the two B[*a*]P DNA adducts. Unlike the B[*a*]P adducts, the photolesions diminish the unwinding rates (*k*_obs_) only by a factor of ~2 for CPD and ~5 for the 6–4PP lesion relative to the unmodified DNA (Table 1). However, they exhibit great impacts on their processivity (*P* = 0.11 for CPD, *P* = 0.27 for 6–4PP); in particular, the *P* value of the CPD-containing duplex is reduced greatly by ~90%, indicating that full unwinding is scarcely achieved (Table 1).

These strong reductions in *P* values are due not only to the reduction in the unwinding rate constant *k*_U_ but also to a strong enhancement of the dissociation rate constant *k_D_*. An NMR study indicates that Watson–Crick base pairings are absent at the 6–4PP lesion site and at the adjacent base pair on its 3’-side, and the duplex is bent by ~44°, which is likely to destabilize the local double-stranded DNA structure. By contrast, the Watson–Crick pairing at the CPD lesion site is more or less maintained, with a much smaller bend in the duplex of only ~9° [30,31]. Although these structural features revealed that the 6–4PP lesion is more distorting than the CPD, their unwinding rate constants *k*_U_ are very similar, within experimental error (Figure 7). Thus, the key kinetic parameter governing the processivity is the dissociation rate constant *k*_D_, which, in the case of the CPD–duplex, is much greater than it is in the case of the 6–4PP–duplex. 

Our previous molecular dynamics (MD) study of the eukaryotic 5′ → 3′ helicase XPD, which plays a key role and acts as a main lesion sensor in the human nucleotide excision repair (NER) mechanism [44], showed that XPD-bound single-stranded DNA molecules containing a CPD or 6–4PP lesion outside the XPD pore are treated differently [41]. The open-book-like thymines of the CPD are sterically blocked from entry; however, the near-perpendicular thymines of 6–4PP can enter and are tightly bound by intermolecular interactions within the pore. The XPD helicase traps the 6–4PP lesion via the displacement of the XPD Arch domain toward the lesion, providing stronger local helicase–lesion interactions, which are not seen in the CPD case. These different behaviors of the two crosslinked thymine dimers are also observed in MD simulations of Nsp13 bound to ssDNA (manuscript in preparation). This may explain why Nsp13 dissociates from the helicase–CPD complex faster than from the helicase–6–4PP complex (Figure 7). 

Notably, these UV-radiation-induced crosslinked thymine dimers exhibit much faster dissociation rate constants compared to B[*a*]P DNA adducts (Figure 7). The B[*a*]P DNA adducts have multi-polycyclic aromatic ring systems that interact with the helicase via non-covalent Van der Waals interactions; however, such interactions are diminished for the UV-induced lesions due to their non-aromatic nature. These interaction differences may account for the faster *k*_D_ rates in photolesions, suggesting that the Nsp13 helicase dissociates faster from the duplex containing photolesions than from the B[*a*]P-lesion-containing duplexes.

### 3.4. The Impact of the Same Set of DNA Lesions on Unwinding Kinetics in Two Different Helicases: SF1 Helicase Nsp13 and SF2 Helicase RecQ

A previous study of the unwinding characteristics of the same set of DNA lesions by *E. coli* RecQ helicase, an SF2 helicase with a 3′ → 5′ polarity, exhibited some similarities to and some differences from the present Nsp13 study [28]. The similarities include that the strongest inhibition of the unwinding activities of both helicases is caused by the UV-induced lesions, while they are less inhibited by the polycyclic aromatic DNA adducts. There are also some differences in the inhibitory effects of the DNA lesions on these two helicases. The duplex containing a B[*a*]P DNA adduct can be fully unwound by RecQ helicase, eventually reaching *P* = 1, regardless of the stereochemistries of the B[*a*]P-dG:dC adducts. However, the duplex is only partially unwound by the Nsp13 helicase and in a stereospecific manner, as reflected in their processivities *P*(*trans*) = 0.89 and *P*(*cis*) = 0.33, indicating that Nsp13 unwinding is sensitive to adduct stereochemistry, and it is possible that Nsp13 might be sensitive to stereochemistry in the case of other DNA lesions. Furthermore, profound inhibition of the human 3′ → 5′ WRN helicase activity was observed for these B[*a*]P-derived dG adducts in the Brosh laboratory [45]. WRN-catalyzed duplex unwinding was strongly inhibited by both *trans* and *cis* adducts situated in the helicase-translocating strand; however, the extent of helicase inhibition was not significantly dependent on the stereochemistry or orientation of the adducts.

For the UV-induced lesions, the helicases RecQ and Nsp13 handle the CPD lesion in a very similar way; the unwinding rate constants (*k*_obs_) are decreased by ~50% relative to the unmodified DNA and the CPD processivities (*P* = 0.11) are decreased by ~90% in both helicases. However, their unwinding activities respond very differently to the 6–4PP lesion, especially in their processivities, with *P* = 0.27 for Nsp13 and a particularly low *P* value of 0.062 for RecQ; this indicates that RecQ holds the 6–4PP lesion less tightly so that it rapidly dissociates from the helicase–duplex complex. These results suggest that RecQ is more highly sensitive to inhibition by the 6–4PP lesion than Nsp13 helicase, whereas the CPD lesion exhibits similar inhibitory effects on both RecQ and Nsp13 helicases. These results address potential differences between the SF2 helicase RecQ and the SF1 helicase Nsp13 in a functional context when they encounter DNA damage. Thus, these comparisons highlight the differential mechanisms of DNA unwinding catalyzed by the different helicases and how specific covalent DNA adducts impact helicase function differently. Different helicase behavior toward different DNA lesions has been demonstrated in our previous MD studies [41] involving the eukaryotic XPD helicase, which acts as the main sensor for verifying the presence of lesions in nucleotide excision repair [44].

## 4. Materials and Methods

The expression of N-terminally His_6_-tagged Nsp13 was conducted as previously described (pET28a 6xHis PreScission SARS-CoV-2 Nsp13 plasmid) [46,47]. Briefly, the pET28a Nsp13 plasmid was transformed into E. coli Rosetta (DE3) cells (Novagen, Madison, WI, USA) and used to inoculate 5 mL of LB medium supplemented with kanamycin (25 μg/mL) and chloramphenicol (30 μg/mL) prior to shaking overnight (37 °C). The next day, cultures were used to inoculate 500 mL of LB supplemented with the same antibiotics and grown to the log phase (OD_600_ = 0.6, optical density at 600 nm) with shaking (37 °C, 200 RPM). Cultures were then induced with 0.2 mM isopropyl β-D-1-thiogalactopyranoside (IPTG) for 17 h with shaking (16 °C). The cells were pelleted and later resuspended with 15 mL of resuspension buffer (50 mM 2-(4-(2-hydroxyethyl)-1-piperazinyl)-ethanesulfonic acid (HEPES), pH 7.5, 500 mM NaCl, 4 mM MgCl_2_, 5% (*v*/*v*) glycerol, 20 mM imidazole, 5 mM beta-mercaptoethanol (BME), 1 mM ATP, 1 mM phenylmethylsulfonyl fluoride (PMSF)) with protease inhibitors (SIGMAFAST Protease Inhibitor Cocktail Tablets, EDTA-Free). Additionally, lysozyme (0.1 mg/mL) and Dnase I (6.6 μg/mL) were added to the resuspended cells. The cells were rocked at 4 °C for 1 h and then lysed by sonication with an amplitude of 50% for 10 min (30 s on and 30 s off). Samples were then centrifuged (8736× *g*, 30 min, 4 °C), and the resulting supernatant was incubated with 4 mL of washed Ni-NTA resin (Qiagen, Germantown, MD, USA, for 30 min while rocking at 4 °C. The column was washed with 2 × 20 mL of Buffer A (20 mM tris(hydroxymethyl)aminomethane-HCl (Tris-HCl), pH 8.0, 300 mM NaCl, 5% (*v*/*v*) glycerol, 1 mM BME, 1 mM ATP) supplemented with 20 mM imidazole and 2 × 20 mL of Buffer A supplemented with 30 mM imidazole. Samples were eluted with 3 × 15 mL of elution buffer (Buffer A supplemented with 250 mM imidazole). Elution fractions were combined and dialyzed in 2L of dialysis buffer (50 mM HEPES-NaOH pH 7.5, 500 mM NaCl, 4 mM MgCl_2_, 5% (*v*/*v*) glycerol, 20 mM imidazole, 5 mM BME) overnight at 4 °C using SnakeSkin Dialysis Tubing (Thermo Scientific, Waltham, MA, USA, 10K MWCO). Following dialysis, the sample was concentrated and purified by gel filtration with a HiLoad 16/600 Superdex 200 prep grade column (Cytiva, Marlborough, MA, USA) in 25 mM HEPES-NaOH pH 7.5, 250 mM KCl, 1 mM MgCl_2_, and 1 mM tris(2-carboxyethyl) phosphine (TCEP) using an ÄKTA FPLC (Cytiva, Marlborough, MA) at a 0.3 mL/min flow rate over 1 column volume. Purified Nsp13 was put into storage buffer (25 mM HEPES-NaOH pH 7.5, 250 mM KCl, 1 mM MgCl_2_, 20% (*v*/*v*) glycerol, 1 mM TCEP) and concentrated. The protein concentration was determined using a DC assay (Bio-Rad, Hercules, CA, USA). Samples were then aliquoted, flash-frozen with liquid N_2_, and stored at −80 °C. 

The CPD and 6–4PP photolesions were generated by UV irradiation of the oligonucleotide 5′-d(GCAAGTTGGAG) in aqueous solutions. The oligonucleotide sequences containing the different photolesions were separated from one another by HPLC methods and further purified by gel electrophoresis. The two B[*a*]P DNA adducts were derived from the *cis- or trans-*addition of the exocyclic amino group of guanine to the C10 atom of the aromatic diol epoxide enantiomer ((+)-7*R*,8*S*)-dihydroxy-(9*S*,10*R*)-epoxy-7,8,9,10-tetrahydrobenzo[*a*]pyrene ((+)-*anti*-B[*a*]P), with the oligonucleotide 5′-d(CCATCXCTACC) with X = dG. The oligonucleotide substrates (Integrated DNA Technologies) used in the unwinding assays are shown in Figure 2.

The sequences displayed in Figure 2 are referred to as single-strand/double-strand (ss/ds) DNA in this work. For each sequence, the top strand contains 25 base pairs (bp) in the double-stranded region and 25 nucleotides (nt) in the single-stranded overhang. The B[*a*]P lesions and the thymine dimers, denoted by X and TT, respectively, are positioned in the double-stranded region of the upper strand, which also contains the black hole quencher (BHQ2) at its 3′-end. The 5′-end of the bottom strand contains the fluorescent Cy3 dye opposite BHQ2. When the double-stranded region is intact, the Cy3 fluorophore is fully quenched by BHQ2 on the opposite strand but reappears when the double-stranded region is unwound by the action of the Nsp13 helicase. The fluorescence emission of Cy3 (λ_max_ = 564 nm) was generated by excitation with a green diode laser (515 nm), and the fluorescence was monitored via a home-built photomultiplier–amplifier system; the signal output was digitized and stored in a computer for further analysis.

## 5. Summary and Conclusions

In the present study, the lesion-dependent kinetic parameters highlight the differing impacts of structurally distinct DNA lesions on the unwinding of double-stranded DNA catalyzed by the Nsp13 helicase. Hence, our findings indicate that the unwinding efficiencies are lesion-dependent and most strongly inhibited by the CPD lesion; moreover, processivity is a better measure of DNA lesions’ inhibitory effects than the unwinding rate constant. The characterization of the effects of structurally and stereochemically different DNA adducts on unwinding catalyzed by Nsp13 helicase presented in this work may be insightful for understanding the mechanistic aspects of helicase function and its possible inhibition for drug design.

## Figures and Tables

**Figure 3 ijms-25-07930-f003:**
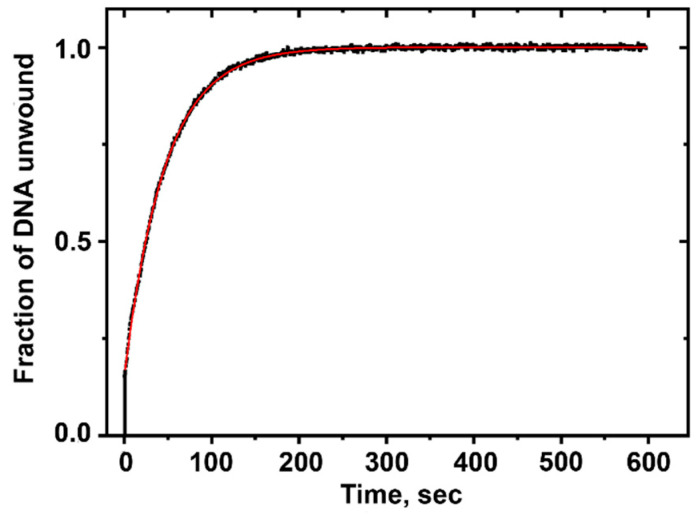
The unwinding kinetics of unmodified DNA substrates (5 nM) catalyzed by Nsp13 (60 nM). The black bar denotes the initial fluorescence burst associated with the Nsp13 helicase bound to the DNA. The curve in red represents Equation (1) with *k*_obs_ = (21.9 ± 0.03) × 10^−3^ s^−1^.

**Figure 4 ijms-25-07930-f004:**
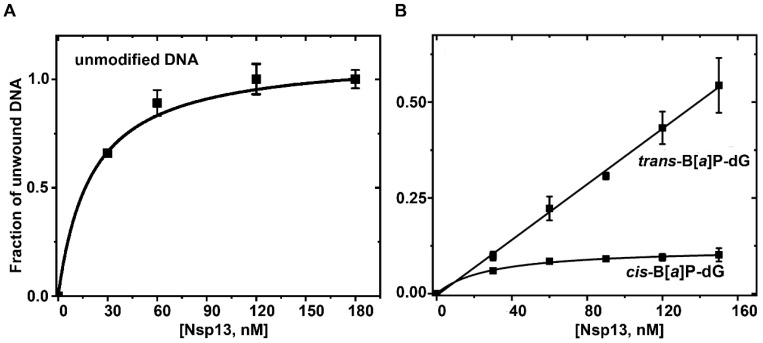
The dependence of the extent of DNA unwinding on the Nsp13 helicase concentration. (**A**) Unmodified DNA, burst (pre-bound DNA) phase only, relative to the amplitude recorded at 180 nM. (**B**) Relative rates for the same sequences but containing a single B[*a*]P-dG adduct. The data points represent the means of at least three independent experiments with standard deviations indicated by the error bars. The error bars are not larger than the sizes of the data points, unless indicated otherwise. In this set of experiments, bursts were not observed.

**Figure 5 ijms-25-07930-f005:**
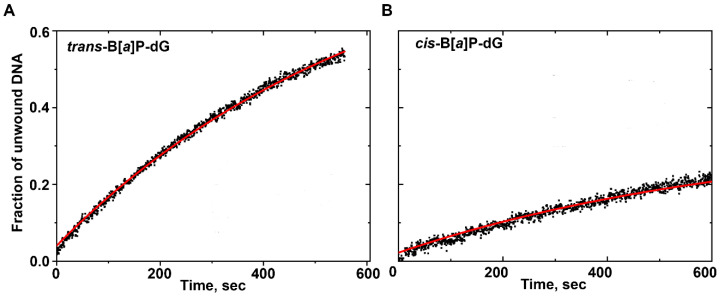
DNA unwinding curves for (**A**) *trans*-B[*a*]P-dG and (**B**) *cis*-B[*a*]P-dG adducts. The curve in red represents the best fit of Equation (1) to the experimental data points (black dots).

**Figure 6 ijms-25-07930-f006:**
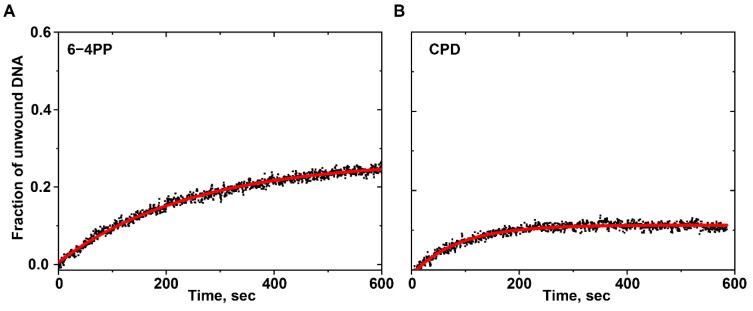
DNA unwinding curves for (**A**) 6–4PP and (**B**) CPD lesions. The curve in red represents the best fit of Equation (1) to the experimental data points (black dots).

**Figure 7 ijms-25-07930-f007:**
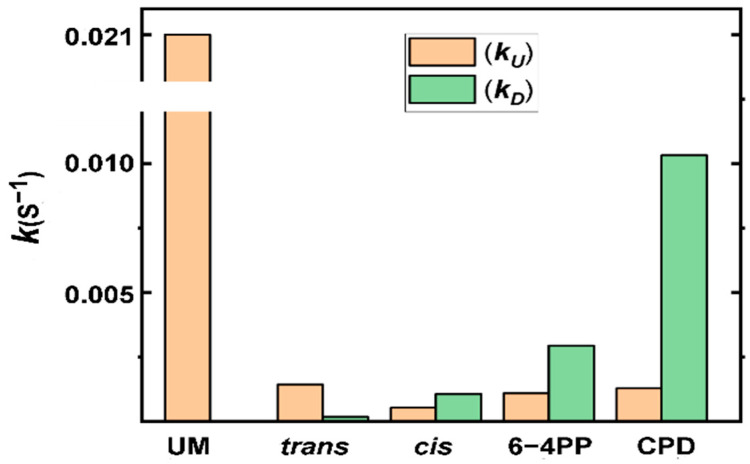
The impact of single bulky B[*a*]P adducts or UV-induced lesions, positioned in the translocating DNA strand, on the rate constants (*k*_U_ = unwinding rate constant; *k*_D_ = helicase–DNA dissociation rate constant). UM refers to the case of unmodified DNA, in which the *k*_D_ rate constant was too slow to be estimated.

**Table 1 ijms-25-07930-t001:** A summary of the overall unwinding rate constants (*k*_obs_), *k*_D_*/k*_U_ ratios, and processivities (*P*) associated with the different DNA systems studied.

DNA Lesion	Processivity (*P*)	*k* _D_ */k* _U_	*k*_obs_ (=*k*_U_ *+ k*_D_), s^−1^
Unmodified DNA	1.00	<<0.10	(21.90 ± 0.03) × 10^−3^
*trans*-B[*a*]P-dG	0.89	0.12	(1.60 ± 0.02) × 10^−3^
*cis*-B[*a*]P-dG	0.33	2.13	(1.47 ± 0.03) × 10^−3^
6–4PP	0.27	2.72	(4.02 ± 0.04) × 10^−3^
CPD	0.11	7.20	(11.60 ± 0.04) × 10^−3^

## Data Availability

Data is contained within the article.

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
