# Peer review of "Variable Inhibition of DNA Unwinding Rates Catalyzed by the SARS-CoV-2 Helicase Nsp13 by Structurally Distinct Single DNA Lesions"

_ijms, 2024, doi:10.3390/ijms25147930_

Round 1

Reviewer 1 Report

Comments and Suggestions for Authors

The authors present an interesting and timely analysis of the biochemical properties of the SARS-CoV-2 helicase nsp13 in the presence of DNA lesions. The data generally support the conclusions of the paper, but additional information and clarification are needed regarding the model that is proposed to explain the data.

It would be helpful to some readers to provide a definition of “bulky DNA adducts” since some would include 6,4-PP and CPDs in this category.

A schematic illustrating the assay concept (similar to Fig 2. In ref 22) would be helpful for illustrating the orientation of the two strands, the dye and quencher, the lesions, and the helicase.

Where error bars are shown, information about the number of replicates and statistical analysis should be provided

The figure numbering is completely off and needs to be corrected

Line 246: what does “both adducts” refer to? Presumably the cis and trans B[a]P adducts, but needs to be stated explicitly

Section 4.2 is wordy and somewhat repetitive; the points made at the beginning of the last paragraph are already made, at least implicitly, in the preceding paragraphs.

Line 303: “their processivities” should be “processivity” 

Line 347: it should be clarified that nsp13 appears to be insensitive to B[a]P adduct stereochemistry in particular; nsp13 might be sensitive to stereochemistry for other DNA lesions

Since there is room in table 1, it would be informative to report the estimated values for kD and kU.

In the plot labeled figure 3 (with the broken y-axis), no kD value is visible for unmodified DNA. Taken together with the upper limit estimate (<<0.1) for kD/kU in Table 1, it appears the rate constant was too slow to be estimated; is this so? It should be stated explicitly in the figure.

The proposed model to explain the data and the underlying assumptions need to be illustrated and articulated more clearly. As defined, processivity refers to the probability of complete unwinding, but it is not clear weather P = 0.1, for example, would mean that 10% of the duplexes are completely unwound or rather that the duplexes are collectively unwound by an average of 10%. Is it assumed that when the helicase dissociates from a partially unwound substrate, a second helicase molecule or the same helicase molecule re-binding cannot re-initiate unwinding? And if not, how would a plateau in the fraction of unwound DNA arise (e.g. at a fraction of approximately 0.1 for CPD)? If a 6-4PP can trap nsp13 (analogous to the situation with XPD), how would this affect estimates of kU, kD, and P?

Comments on the Quality of English Language

The manuscript reads well but would benefit from a careful round of proofreading to clarify some sections that could be interpreted in more than one way.

Author Response

Please note that the changes in the revised manuscript are highlighted in red font.

The authors present an interesting and timely analysis of the biochemical properties of the SARS-CoV-2 helicase nsp13 in the presence of DNA lesions. The data generally support the conclusions of the paper, but additional information and clarification are needed regarding the model that is proposed to explain the data. 

Comments 1: It would be helpful to some readers to provide a definition of “bulky DNA adducts” since some would include 6,4-PP and CPDs in this category.

Response 1: Thank you for pointing this out. The designation ‘bulky’ refers to the physical dimensions of the B[a]P aromatic ring system. We have provided the definition of bulky DNA adducts in the manuscript.

Page 2, line 47:

“The designation ‘bulky’ refers to the physical dimensions of the B[a]P aromatic ring system”

And page 2 and lines 73-74:

“The B[a]P-derived DNA adducts are designated as ‘bulky’ because they contain five hexagonal carbon rings, while the non-bulky CPD and 6-4 contain none”

Comment 2: A schematic illustrating the assay concept (similar to Fig 2. In ref 22) would be helpful for illustrating the orientation of the two strands, the dye and quencher, the lesions, and the helicase.

Response 2: We agree! We have provided a new Figure 2 (Definitions of the DNA substrates), which is similar to Figure 2 in our prior study of the RecQ helicase [28]. See page 3.

[28] A.H. Sales, V. Zheng, M.A. Kenawy, M. Kakembo, L. Zhang, V. Shafirovich, S. Broyde, N.E. Geacintov, Inhibition of E. coli RecQ Helicase Activity by Structurally Distinct DNA Lesions: Structure-Function Relationships, Int J Mol Sci 23 (2022). 

Comment 3: Where error bars are shown, information about the number of replicates and statistical analysis should be provided.

Response 3: Thank you for this comment. All error bars are based on the averages of at least three measurements. We have now added information about replicates for the experiments in the manuscript. See the caption of Figure 4, page 5.

Figure 4. Dependence of extent of DNA unwinding on the Nsp13 helicase concentration. (A) Unmodified DNA, burst (pre-bound DNA) phase only, relative to the amplitude recorded at 180 nM. (B) Relative rates, same sequences, but containing a single B[a]P-dG adduct. The data points represent the mean of at least three independent experiments with standard deviations indicated by the error bars. The error bars were not larger than the sizes of the data points, unless indicated otherwise. In this set of experiments, bursts were not observed.”  

Comment 4: The figure numbering is completely off and needs to be corrected

Response 4: The errors in the figure numbering are editorial errors by the publishing program. Our original manuscript was accurate! We have restored the manuscript to the original accurate form.  

Comment 5: Line 246: what does “both adducts” refer to? Presumably the cis and trans B[a]P adducts, but needs to be stated explicitly

Response 5: Thank you for carefully pointing this out. We have changed “both adducts” to “both trans- and cis-B[a]P adducts” in the revised manuscript.

See page 8, lines 243-245.:

“The presence of the B[a]P-DNA lesions in the double-stranded region of the modified DNA sequences (Figure 1B and Figure 2) results in a significant decrease in the overall unwinding rate constant kobs to a similar extent for both trans and cis adducts,…..”

Comment 6: Section 4.2 is wordy and somewhat repetitive; the points made at the beginning of the last paragraph are already made, at least implicitly, in the preceding paragraphs.

Response 6: We have shortened the first paragraph of this section, which summarizes the results to make it more focused. The 2nd and 3rd paragraphs are interpretations of the results, which are necessary and appropriate for the Discussion.

See Section 4.2. on page 8 and lines 243-249:    

“The presence of the B[a]P-DNA lesions in the double-stranded region of the modified DNA sequences (Figure 1B and Figure 2) results in a significant decrease in the overall unwinding rate constant kobs to a similar extent for both trans and cis adducts, but there is a pronounced difference in their processivities (Table 1). These observations indicate that when compared to unmodified DNA, for the trans case, the decrease in kobs is predominantly due to the decrease in the unwinding rate constant kU (Figure 7), which is likely associated with steric hindrance effects. However, for the cis case, the decrease in kobs is caused not only by a much slower unwinding rate constant kU, but also by a much faster dissociation rate constant kD than in the trans adduct (Figure 7).” 

Comment 7: Line 303: “their processivities” should be “processivity”

Response 7: Corrected. See page 9 and line 283.  

Comment 8: Line 347: it should be clarified that nsp13 appears to be insensitive to B[a]P adduct stereochemistry in particular; nsp13 might be sensitive to stereochemistry for other DNA lesions.

Response 8: Thanks for the comment. We have revised accordingly to emphasize this point. See page 9 and lines 316-319:

“However, the duplex is only partially unwound by the Nsp13 helicase and in a stereospecific manner, as reflected in their processivities P(trans) = 0.89 and P(cis) = 0.33, indicating that Nsp13 unwinding is sensitive to adduct stereochemistry, and it is possible that Nsp13 might be sensitive to stereochemistry in the case of other DNA lesions.”

Comment 9: Since there is room in table 1, it would be informative to report the estimated values for kD and kU.

Response 9: However, since the data of kU and kD are depicted clearly in Figure 7, adding them to Table 1 does not provide any additional information.

Comment 10: In the plot labeled figure 3 (with the broken y-axis), no kD value is visible for unmodified DNA. Taken together with the upper limit estimate (<<0.1) for kD/kU in Table 1, it appears the rate constant was too slow to be estimated; is this so? It should be stated explicitly in the figure.

Response 10: We agree. We have explicitly stated that the rate constant kD was too slow to be estimated for the unmodified DNA unwinding.

In the Figure 7 caption on page 7,

Figure 7. Impact of single bulky B[a]P adducts or UV-induced lesions, positioned in the translocating DNA strand, on the rate constants. (kU = unwinding rate constant; kD = helicase–DNA dissociation rate constant). UM refers to the case of unmodified DNA, in which the kD rate constant was too slow to be estimated.”

Comment 11: The proposed model to explain the data and the underlying assumptions need to be illustrated and articulated more clearly. As defined, processivity refers to the probability of complete unwinding, but it is not clear whether P = 0.1, for example, would mean that 10% of the duplexes are completely unwound or rather that the duplexes are collectively unwound by an average of 10%. Is it assumed that when the helicase dissociates from a partially unwound substrate, a second helicase molecule or the same helicase molecule re-binding cannot re-initiate unwinding? And if not, how would a plateau in the fraction of unwound DNA arise (e.g. at a fraction of approximately 0.1 for CPD)? If a 6-4PP can trap nsp13 (analogous to the situation with XPD), how would this affect estimates of kU, kD, and P?

Response 11: Here we provide answers to the reviewer’s comments.

In the case of CPD, the P value = 0.1 indicates that 10 % of the DNA duplexes are completely unwound. It would take the helicase ~ 200 seconds to fully unwind the duplex before it disassociates from the DNA as shown in Figure 5B, explaining a rise in the fraction of unwound DNA from 0 to 200 seconds before it reaches the plateau. For the 6−4PP case, with the value P = 0.27, which is approximately 2.5 times greater than the one in CPD, which is mainly due to the slower disassociation rate constant of helicase from 6−4PP than CPD (as their kU values are very similar (see Figure 7)). Our MD simulations of XPD and Nsp13 complexed with DNA containing a UV-induced lesion right outside the tunnel have revealed that only the 6−4PP lesion can enter and be trapped within the tunnels of both helicases. Thus, this trapped 6−4PP forms stronger interactions with the helicase than those in the case of CPD, explaining why kD in 6−4PP is much smaller than in CPD, which results in a much higher value of P than in the case of CPD.

These points have been made in the manuscript on page 9 and lines 294-302.   

Comment 12: Comments on the Quality of English Language

The manuscript reads well but would benefit from a careful round of proofreading to clarify some sections that could be interpreted in more than one way

Response 12: We have carefully proofread and edited the manuscript for clarity.

Reviewer 2 Report

Comments and Suggestions for Authors

Review of the manuscript IJMS-3049124

This aim of this work is very interesting.

It is well-known that an imbalance between the oxidant and antioxidant mechanisms after exposure to deleterious stimuli causes oxidative stress leading to increased production of superoxide anion radical and hydrogen peroxide. Oxidative stress plays a pivotal role in the pathogenesis of viral infections. Acute immune activation following a viral infection is associated with increased oxidative stress, as a result of viral replication and the consequent inflammation. Under these inflammatory conditions, the immune cellular components generate increased levels of reactive oxygen (ROS) and nitrogen species (RNS), two main oxidative representatives, and catalyze the production of oxidative DNA damage and the activation of DNA Damage Response (DDR), indicating an association among acute immune activation, augmented oxidative stress and DNA damage formation.

 COVID-19 is an infectious disease caused by the virus known as severe acute respiratory syndrome coronavirus 2 (SARS-CoV-2; a single-stranded RNA virus). It is well-established that severe oxidative stress in COVID-19 is linked to the severe phase of COVID-19 and this is associated with acute inflammation in lungs, greater loss of lung perfusion, severe hypoxic vasoconstriction, and hypoxemia mediated by robust secretion of cytokines (“cytokine storm”) and chemokines by immune cells (particularly macrophages and Th1 cells).

Now, in this work, the authors used two non-bulky DNA-lesions - cyclobutene dimers and 64PP, as test or example cases although they are not relevant for the COVID 19 until and unless the authors are using these lesions to characterize DNA-based COVID 19 vaccines.

The two other bulky lesions considered are the benzo[a]pyrene (B[a]P)-modified guanine: cytosine lesions. The point is that the logic of choosing the two non-bulky and two bulky DNA-lesions and these lesions are not clear. Thus, the presentation of this work lacks providing the correct “take home message” to the readers of this manuscript.

The obvious question is why this study is important from the perspective of COVID-19. The methodology adopted in this work, the results and discussion are fine.

Once the authors address these issues, I would like it to review it.

Author Response

Please note that the changes in the revised manuscript are displayed in red font.

===

This aim of this work is very interesting.

It is well-known that an imbalance between the oxidant and antioxidant mechanisms after exposure to deleterious stimuli causes oxidative stress leading to increased production of superoxide anion radical and hydrogen peroxide. Oxidative stress plays a pivotal role in the pathogenesis of viral infections. Acute immune activation following a viral infection is associated with increased oxidative stress, as a result of viral replication and the consequent inflammation. Under these inflammatory conditions, the immune cellular components generate increased levels of reactive oxygen (ROS) and nitrogen species (RNS), two main oxidative representatives, and catalyze the production of oxidative DNA damage and the activation of DNA Damage Response (DDR), indicating an association among acute immune activation, augmented oxidative stress and DNA damage formation.

COVID-19 is an infectious disease caused by the virus known as severe acute respiratory syndrome coronavirus 2 (SARS-CoV-2; a single-stranded RNA virus). It is well-established that severe oxidative stress in COVID-19 is linked to the severe phase of COVID-19 and this is associated with acute inflammation in lungs, greater loss of lung perfusion, severe hypoxic vasoconstriction, and hypoxemia mediated by robust secretion of cytokines (“cytokine storm”) and chemokines by immune cells (particularly macrophages and Th1 cells).

Comment 1: Now, in this work, the authors used two non-bulky DNA-lesions - cyclobutene dimers and 64PP, as test or example cases although they are not relevant for the COVID 19 until and unless the authors are using these lesions to characterize DNA-based COVID 19 vaccines.

The two other bulky lesions considered are the benzo[a]pyrene (B[a]P)-modified guanine: cytosine lesions. The point is that the logic of choosing the two non-bulky and two bulky DNA-lesions and these lesions are not clear. Thus, the presentation of this work lacks providing the correct “take home message” to the readers of this manuscript.

Response 1: Thank you for pointing this out. We have, accordingly, clarified the logic of choosing the two UV-induced DNA lesions and the two B[a]P adducts in the revised manuscript. Page 2, lines 43-48:

“The objective of this work was to gain a new understanding of how the chemical structures and physical conformations of different DNA lesions (Figure 1) affect DNA unwinding catalyzed by the Nsp13 helicase. We focused on the Nsp13-catalyzed unwinding of double-stranded DNA containing single UV-induced DNA photolesion and single benzo[a]pyrene-diol epoxide (B[a]P) reaction products that in vivo or in vitro form the bulky B[a]P-N2-dG (guanine) adducts shown in Figure 1. The designation ‘bulky’ refers to the physical dimensions of the B[a]P aromatic ring system. These adducts are known from previous studies to inhibit helicases [27-29] including Nsp13.”

Comment 2: The obvious question is why this study is important from the perspective of COVID-19.

Response 2: The methodology presented in this work demonstrated how the chemical structures of different DNA lesions and their conformations in double-stranded DNA affect the DNA unwinding performed by the nsp13 helicase in order to elucidate their impacts on the mechanistic aspects of the helicase nsp13 function, which plays a critical role in the viral replication and proliferation . We have specified the importance of this study from the perspective of COVID-19. See page 1, lines 25-30:

“The non-structural protein 13 (Nsp13), an RNA helicase, belongs to the helicase superfamily 1 (SF1) and plays a critical role in the viral replication; it first unwinds double-stranded RNA to provide a single-stranded template for the subsequent transcription of the viral genome. The COVID19 pandemic stimulated significant interest in the design of new inhibitors [3-9] to suppress the SARS-CoV-2 Nsp13 helicase unwinding activities and subsequent viral replication and proliferation [10, 11].”  

The methodology adopted in this work, the results and discussion are fine.

Once the authors address these issues, I would like it to review it.

Reviewer 3 Report

Comments and Suggestions for Authors

The authors are examining the effect of DNA lesions on the DNA unwinding by the SARS-CoV-2 nsp13 helicase.

Any initial reading of a paper begins with the abstract and examination of the figures and conclusion. This requires that the figure captions be self explanatory. Please specify in the Figure 3 legend that the data are for the burst (pre-bound DNA) phase only.

This analysis used a 5nM dsDNA solution with 60nM of helicase, which is lower than the typical DNA unwinding assay which uses about 250 nM of DNA. The use of an assay in which the enzyme outnumbers the substrate seems odd. Can you explain your reasoning beyond the low DNA recombination rate? In addition, although the unwinding buffer is mentioned, there is no detailed explanation of its contents or the ATP concentration used.

Please compare the effects of these legions on nsp13 with their effect on other helicases, particularly host helicases not just your previous work on E. coli RecQ . An interesting side note would be to compare these lesions with the inhibition of nsp13 by SARS/MERS specific inhibitors (https://covdb.stanford.edu/search/?target=Helicase).

The conclusions state that the effects of DNA lesions on nsp13 DNA unwinding may point to the design of inhibitors. This is an overly broad statement. Unless, a DNA lesion’s effect is specific to nsp13 and does not effect any host helicases then it does not present a viable candidate for inhibitor design. Unless you can present an example of DNA mutation inducing or mimicking drugs used to treat a disease other than cancer then this statement cannot be supported.

The search for inhibitors of SARS nsp13 is a very active area of current research. However, there are no references to any such work beyond 2021. Please update your references to include more recent developments.

Author Response

Please note that the changes in the revised manuscript are displayed in red font. 

====

The authors are examining the effect of DNA lesions on the DNA unwinding by the SARS-CoV-2 nsp13 helicase. 

Comment 1: Any initial reading of a paper begins with the abstract and examination of the figures and conclusion. This requires that the figure captions be self explanatory. Please specify in the Figure 3 legend that the data are for the burst (pre-bound DNA) phase only.

Response 1: Thank you for pointing this out. We have, accordingly, specified in the Figure 4 (It was Figure 3 in the original submitted manuscript and now it is Figure 4 in the revised manuscript) legend that the DNA unwinding data are only for the burst phase (pre-bound DNA) in the revised manuscript.

See page 5, the Figure 4 caption:

Figure 4. Dependence of extent of DNA unwinding on the Nsp13 helicase concentration. (A) Unmodified DNA, burst (pre-bound DNA) phase only, relative to the amplitude recorded at 180 nM. (B) Relative rates, same sequences, but containing a single B[a]P-dG adduct. The data points represent the mean of at least three independent experiments with standard deviations indicated by the error bars. The error bars were not larger than the sizes of the data points, unless indicated otherwise. In this set of experiments, bursts were not observed”

Comment 2: This analysis used a 5 nM dsDNA solution with 60 nM of helicase, which is lower than the typical DNA unwinding assay which uses about 250 nM of DNA. The use of an assay in which the enzyme outnumbers the substrate seems odd. Can you explain your reasoning beyond the low DNA recombination rate?

Response 2: Thank you for pointing this out. The cost of fabricating the site-specifically modified oligonucleotides. The fabrication of the high-purity DNA substrates containing a single modified DNA base is very time consuming and expensive. The low 5 nM DNA concentration has therefore been adopted since it yields excellent reproducible signals and we have enough substrates for multiple experiments.

Comment 3: In addition, although the unwinding buffer is mentioned, there is no detailed explanation of its contents, or the ATP concentration used.

Response 3: We have provided the requested information.

See page 4 and lines 129-133:

“Prior to the unwinding experiments, the DNA substrates were pre-equilibrated with the Nsp13 protein (60 nM), and the unwinding reactions were initiated by mixing this solution with aliquots of the unwinding buffer, containing 20 mM TRIS-HCl, pH 7.6, 10 mM KCl, 5 mM MgCl2, 2 mM dithiothreitol (DTT), 5% glycerol, 0.1 μg/μL bovine serum albumin (BSA), and 2mM ATP. All experiments were conducted at 25 °C.”

Comment 4: Please compare the effects of these lesions on nsp13 with their effect on other helicases, particularly host helicases not just your previous work on E. coli RecQ.

Response 4: Thank you for bringing up this important matter. We have now considered the impact of B[a]P-derived adducts on the 3’ à 5’ WRN helicase.

See page 9, lines 319-323:

“Furthermore, a profound inhibition of the human 3’ à 5’ WRN helicase activity was observed for these B[a]P-derived dG adducts in the Brosh laboratory [47]. The WRN-catalyzed duplex unwinding was strongly inhibited by both trans and cis adducts situated in the helicase-translocating strand; however, the extent of helicase inhibition was not significantly dependent on stereochemistry or orientation of the adducts.”

[47] S. Choudhary, K.M. Doherty, C.J. Handy, J.M. Sayer, H. Yagi, D.M. Jerina, R.M. Brosh, Jr., Inhibition of Werner syndrome helicase activity by benzo[a]pyrene diol epoxide adducts can be overcome by replication protein A, J Biol Chem 281 (2006) 6000-6009

Comment 5: An interesting side note would be to compare these lesions with the inhibition of nsp13 by SARS/MERS specific inhibitors from  https://covdb.stanford.edu/search/?target=Helicase

Response 5: Thank you for this very interesting suggestion. However, with the many entries in the database it is unfortunately beyond the scope of our current work to make the requested comparisons.  

Comment 6: The conclusions state that the effects of DNA lesions on nsp13 DNA unwinding may point to the design of inhibitors. This is an overly broad statement. Unless, a DNA lesion’s effect is specific to nsp13 and does not effect any host helicases then it does not present a viable candidate for inhibitor design. Unless you can present an example of DNA mutation inducing or mimicking drugs used to treat a disease other than cancer then this statement cannot be supported.

Response 6: We have modified the statement concerning drug design to highlight that our work provides mechanistic understanding of helicase function.

See page 10 and lines 340-343 in Summary and Conclusion,

“Characterization of the effects of structurally and stereochemically different DNA adducts on unwinding catalyzed by Nsp13 helicase presented in this work may be insightful for understanding mechanistic aspects of helicase function, and possible inhibition for drug design.”

Comment 7: The search for inhibitors of SARS nsp13 is a very active area of current research. However, there are no references to any such work beyond 2021. Please update your references to include more recent developments.

Response 7: Thank you for pointing this out. We have updated our references regarding the recently developed inhibitors of SARS nsp13 in the revised manuscript.

See page 1, the 1st paragraph:

“The COVID19 pandemic stimulated significant interest in the design of new inhibitors [3-9] to suppress the SARS-CoV-2 Nsp13 helicase unwinding activities and subsequent viral replication and proliferation [10, 11].”

And page 1, the 2nd paragraph:

“Significant efforts have been invested to identify effective inhibitors of the ATPase activity of nsp13 [4, 6, 8, 11, 22-24].”

Additional references:

[6] N. Mehyar, Coronaviruses SARS-CoV, MERS-CoV, and SARS-CoV-2 helicase inhibitors: a systematic review of invitro studies, J Virus Erad 9 (2023) 100327.

[7] A. Corona, K. Wycisk, C. Talarico, C. Manelfi, J. Milia, R. Cannalire, F. Esposito, P. Gribbon, A. Zaliani, D. Iaconis, A.R. Beccari, V. Summa, M. Nowotny, E. Tramontano, Natural Compounds Inhibit SARS-CoV-2 nsp13 Unwinding and ATPase Enzyme Activities, ACS Pharmacology & Translational Science 5 (2022) 226-239.

[8] A. Corona, V.N. Madia, R. De Santis, C. Manelfi, R. Emmolo, D. Ialongo, E. Patacchini, A. Messore, D. Amatore, G. Faggioni, M. Artico, D. Iaconis, C. Talarico, R. Di Santo, F. Lista, R. Costi, E. Tramontano, Diketo acid inhibitors of nsp13 of SARS-CoV-2 block viral replication, Antiviral Research 217 (2023) 105697.

[9] G. Li, R. Hilgenfeld, R. Whitley, E. De Clercq, Therapeutic strategies for COVID-19: progress and lessons learned, Nature Reviews Drug Discovery 22 (2023) 449-475.

Round 2

Reviewer 1 Report

Comments and Suggestions for Authors

The authors have addressed all of my concerns adequately.

Reviewer 2 Report

Comments and Suggestions for Authors

The authors have addressed nicely to my queries. I recommend "accept".